# Carbodiimide Conjugation of Latent Transforming Growth Factor β1 to Superparamagnetic Iron Oxide Nanoparticles for Remote Activation

**DOI:** 10.3390/ijms20133190

**Published:** 2019-06-29

**Authors:** Obiora Azie, Zachary F. Greenberg, Christopher D. Batich, Jon P. Dobson

**Affiliations:** 1Department of Materials Science & Engineering, University of Florida, Gainesville, FL 32603, USA; 2J Crayton Pruitt Family Department of Biomedical Engineering, University of Florida, Gainesville, FL 32611, USA

**Keywords:** TGF-β, SPION, conjugation, remote activation

## Abstract

Conjugation of latent growth factors to superparamagnetic iron oxide nanoparticles (SPIONs) is potentially useful for magnetically triggered release of bioactive macromolecules. Thus, the goal of this work was to trigger the release of active Transforming Growth-Factor Beta (TGF-β) via magnetic hyperthermia by binding SPIONs to the latent form of TGF-β, since heat has been shown to induce release of TGF-β from the latent complex. Commercially available SPIONS with high specific absorption rates (SAR) were hydrolyzed in 70% ethanol to create surface carboxylic acid conjugation sites for carbodiimide chemistry. Fourier-Transform Infra-Red (FTIR) analysis verified the conversion of maleic anhydride to maleic acid. 1-Ethyl-2-(3-dimethyulaminopropyl) carbodiimide (EDC) and N-hydroxysulfosuccinimide (Sulfo-NHS) were used to bind to the open conjugation sites of the SPION in order to graft latent TGF-β onto the particles. The resulting conjugated particles were imaged with transmission electron microscopy (TEM), and the complexed particles were characterized by dynamic light scattering (DLS) and superconducting quantum interference device (SQUID) magnetometry. Enzyme-linked immunosorbent assay (ELISA) was used to assess the thermally triggered release of active TGF-β from the latent complex, demonstrating that conjugation did not interfere with release. Results showed that latent TGF-β was successfully conjugated to the iron oxide nanoparticles, and magnetically triggered release of active TGF-β was achieved.

## 1. Introduction

Transforming growth factor beta (TGF-β) is a highly studied cytokine that participates in or controls a wide variety of cell signaling pathways. These include control of cell cycle [1], cell proliferation [2,3], extracellular matrix formation [4,5], and even stem cell differentiation [6,7]. However, TGF-β is also implicated in cell cytosis [8], cellular senescence [9], apoptosis [10], and even tumor formation [11]. Due to its potential to induce these potentially dangerous off-target effects, TGF-β is rarely used in a clinical setting [12]. As such, a method to spatially and temporally control the release of TGF-β, preferably in a non-invasive manner, is required to more fully unlock its clinical potential.

Within the body, TGF-β exists sequestered by a latency associated peptide (LAP), forming what is known as the small latent complex (SLC) [13]. The active form of TGF-β can be released from the SLC via a variety of different methods, including mechanical unfolding of the LAP by cell surface integrins, as well as chemical interaction with thrombospndin-1 or reactive oxygen species [14,15]. However, the active form of TGF-β can also be released via a temperature increase or induced shear stress [16]. Lin et al. showed that they were able to successfully conjugate the SLC to carbon nanotubes, via carbodiimide crosslinking chemistry, and then thermally release the active growth factor by irradiating the carbon nanotubes with near-infrared light [17]. However, in vivo applications of this method are limited by the penetration depth of near-infrared light (1–2 cm [18]). Ergo, another method of triggered release combined with greater penetration depth is needed to provide therapeutic effects to deep tissues.

Superparamagnetic iron oxide nanoparticles (SPIONs) [19] have begun to see widespread use in several different biomedical applications [20,21,22], including magnetically targeted drug delivery [23], magnetically induced drug [24] and protein release [25], and magnetic activation of receptor signaling (MARS) [26]. An additional application, magnetic hyperthermia, utilizes a property of SPIONs to couple to an alternating magnetic field, rotate their magnetic moment with the direction of the field, and generate heat as a result of the phase lag [27,28]. Magnetically triggered release of active TGF-β from the SLC offers a potential route to the spatiotemporal control over cell signaling, while also overcoming the problem of penetration depth. In previous work, we have shown magneto-thermal activation of latent TGF-β via sulfosuccinimidyl 4-[N-maleimidomethyl] cyclohexane-1-carboxylate (Sulfo-SMCC) conjugation to SPIONs and exposure to radiofrequency (RF) alternating magnetic fields [29]. This current study aims to replicate the previous work of conjugating latent TGF-β to SPIONS for magneto-thermal activation via the carbodiimide crosslinking chemistry used to conjugate the same protein with carbon nanotubes for near infra-red (NIR) light triggered release. At the same time, this work aims to elucidate the change in material properties of the SPIONS from their pre-conjugated state to their post-conjugated state, as well as give insight into the use of magnetic hyperthermia and the phenomenon of surface heating.

Carbodiimide crosslinking chemistry is used to conjugate a carboxylic acid to a primary amine. Our SPIONS come pre-coated with maleic anhydride; this maleic anhydride can be hydrolyzed into maleic acid, which contains two carboxyl groups. The carboxylic acids on the magnetic nanoparticle and the primary amine of the protein (specifically, on surface lysines and at the N-terminus) are the targets for this carbodiimide conjugation reaction. The process begins with EDC binding to an open carboxyl group to form an unstable o-Acylisourea intermediate [30]. Unfortunately, the rate of hydrolysis of the intermediate back into the carboxyl is much faster than the rate of amide formation, and thus, Sulfo-NHS is typically used to create a dry-stable intermediate [31]. The Sulfo-NHS acts as a nucleophile, which binds to the carbonyl carbon, forming a tetrahedral complex wherein the o-Acylisourea group is substituted by Sulfo-NHS. Another substitution can take place wherein a primary amine can substitute the Sulfo-NHS to form an amide bond. In this case, LAP contains multiple amino acid residues of lysine [32], as well as a terminal amine group—any of which can serve as a possible site for amide bond formation.

## 2. Results

### 2.1. Characterization of Conjugated Particles

#### 2.1.1. Fourier Transform Infrared—Attenuated Total Reflectance Spectroscopy

FTIR analysis showed a peak at ~1700 cm^−1^ that is representative of a carbonyl carbon belonging to a carboxylic acid functional group (Figure 1). When comparing the pre-conjugation particles to the post-conjugation particles, this peak is replaced by the dual peaks at ~1550cm^−1^ and ~1650 cm^−1^, as well as a new peak at ~3300 cm^−1^. The peaks at 1550 and 3300 are mostly representative of an N–H bond found in an amide group. The peak at 1650 is mostly representative of a carbonyl carbon attached to an amine in an amide bond.

Taken together, the loss of the carboxylic acid functional group and the gaining of an amide bond strongly suggest that the conjugation between the magnetic nanoparticle and the protein by way of EDC/NHS, was successful. Furthermore, these results are consistent across multiple different proteins, which suggests that FTIR is a reliable method for testing conjugation chemistry.

#### 2.1.2. Dynamic Light Scattering (DLS)

The bimodal peak distribution in Figure 2c indicates aggregation of the magnetic nanoparticles. The aggregate size was 1 to 3 µm. In Figure 2b, however, it was assumed that the particle size would be akin to that of Figure 2a, around the given size of 30 nm, however, the size distribution of the control group was seemingly an order of magnitude greater than that of the particles alone. The unimodal distribution does not point to aggregation of the particles, but rather to a change in the particle’s hydrodynamic radius. One possible explanation could be the increase of hydrodynamic radius by the ring opening conversion of maleic anhydride to maleic acid, which represents the converted surface species on the magnetic nanoparticles. The additional carboxyl groups are then additionally stabilized and solvated by Phosphate-buffered saline (PBS), which would be the most likely case for the shift in distribution but retention of unimodality.

#### 2.1.3. Transmission Electron Microscopy (TEM)

The TEM images of Figure 3c,d show the formation of a film-like structure surrounding the magnetic nanoparticles after both EDC and NHS were added. If we assume that this film is made of latent TGF-β, then this suggests that the EDC/NHS conjugation reaction effectively crosslinked the nanoparticles. The conversion of maleic anhydride to maleic acid frees up two units of carboxylic acids that are primed for use by carbodiimide conjugation, the density of the surface species, maleic acid, would dictate the degree of binding availability for the protein to be bound to. TGF-β has multiple binding domains available for carbodiimide conjugation chemistry to occur. Thus, the free TGF-β that may have already reacted with one magnetic nanoparticle may yet still react with another magnetic nanoparticle, or even with another protein. This is likely the cause of the aggregation seen in Figure 2c’s DLS results.

#### 2.1.4. Superconducting Quantum Interference Device Magnetometer.

The high magnetization and lack of hysteresis shown in Figure 4A indicate that the particles were superparamagnetic and retained their superparamagnetic properties after conjugation. In Figure 4B, the two groups shared similar volume fractions, as expected. This suggests an accurate Lavange-Chantrell model. Notably, the results shown in Figure 4B also indicate that there was no observable decrease in saturation magnetization caused by the conjugation of the magnetic nanoparticles, though further studies are required to verify the statistical significance of this observation. Interestingly enough, Figure 4B also indicates that the conjugated particles exhibited an increase in magnetic diameter. The magnetic diameter obtained from the Chantrell model often differs from a particle’s physical diameter. Thus, it does not necessarily follow that an increase in physical diameter should also lead to an increase in magnetic diameter, especially when that increase is due to the conjugation of a nominally diamagnetic protein. As such, further investigation into this matter is needed.

### 2.2. ELISA of Activated TGF-β

#### 2.2.1. Magnetic Release of Active TGF-β

Results indicate a statistically significant (*p* < 0.001) increase in the release of active TGF-β from the samples that underwent magnetic hyperthermia, compared to the background release seen in the control. However, the high background release of the control group indicates that either the alternating magnetic field, sample handling, or the carbodiimide crosslinking itself may have led to some release of active TGF-β from the SLC.

#### 2.2.2. Heat Controls

An ELISA specific to active TGF-β was run with samples at room temperature versus samples heated at 55 °C for a period for 1 h, and samples heated at 80 °C for 10 min. As shown by Figure 5a, the bulk temperature of the sample reached as high as 55 °C during magnetic induction. However, heating at 55 °C for a period of 1 h was not enough to cause a significant increase in the release of active TGF-β from the latent complex when compared to the negative control of latent TGF-β left at room temperature, as shown in Figure 5b. Instead, temperature must exceed 80 °C in order to show a significant amount of release.

## 3. Discussion

Previous work on triggered release of active TGF-β, aside from magnetic activation, has focused on the use of carbon nanotubes that are photo-irradiated to activate TGF-β [32] Carbon nanotubes have the ability to generate heat when exposed to ultra-violet (UV) or NIR light. NIR light has the ability to penetrate human tissues with minimal damage. By conjugating carbon nanotubes to the LAP molecule of the SLC, that group was able to achieve remote activation of latent TGF-β. The main problem with NIR light, however, is that its penetration depth is limited to 1–2 cm. Not only does this greatly limit the physiological range wherein such a technique may be used, it also presents further complications when scaling up from small animal experiments. As such, there is a need for an external trigger capable of targeted activation at deep tissue sites.

The results reported here were directed towards the development of a non-invasive technique, similar to carbon nanotubes, to control the release of active TGF-β—and, eventually, other bioactive molecules—in order to minimize deleterious off-target effects. Local injection of active TGF-β at specific sites initiates a cascade of positive and negative effects in many cell types, including unwanted cellular proliferation and tumor growth. Thus, by binding its latent form to SPIONs, we gain the ability to noninvasively release the active form at local sites by focusing fields to specific targets, thereby allowing us to better mitigate and control negative effects. The SPIONs couple to the external field, and allow us to alter the energy landscape of the conjugated protein. Because the human body is largely transparent to magnetic fields and the limits of field strength and frequency are well studied, this technique will potentially give us greater control over cell signaling at most target sites in the body. Carbodiimide chemistry was used to attach latent TGF-β through a zero-linked reaction by reacting open carboxyl groups, by hydrolytic conversion to maleic acid on the SPIONs, to the open primary amine containing side groups. These side groups have been sequenced [33] and are confirmed to have asparagine, glutamine, and arginine. These side chain groups are stabilized into specific tertiary structure/domains and remain an unlikely site for binding, so it is most likely the N-terminus of the protein that is being conjugated. Conjugation to the SPIONs was observed in FTIR-ATR, where the new existing peak of 1650 was observed. 1650 is highly suggestive of covalent carbonyl carbon to a secondary amine.

The magnetic properties of the SPIONS both pre- and post-conjugation were tested in the SQUID magnetometer. Interestingly, the values for saturation magnetization and magnetic diameter were nominally higher for the conjugated particles than they were for the unconjugated particles, however, further studies will be needed to determine the significance of these results. Similarly, both TEM and DLS provided data that seemingly agree with each other: The aggregations that are seen in Figure 2c in DLS were confirmed in Figure 3c/d of TEM. This suggests that there is some amount of crosslinking occurring wherein latent TGF-β binds to multiple NHS-active SPIONs. This could have some effects on the stability of the SLC, and may lead to the high background release seen in the control groups of Figure 6.

In order to test the release of active TGF-β, an ELISA specific to the active form of TGF-β with low cross-reactivity to latent TGF-β was used. It was verified that thermal activation of the latent complex occurred at a temperature of 80 °C, however, neither thermal nor magnetic release appeared to have impacted the structure of the active form of TGF-β, as shown in the ELISA results. Given that the temperature of the conjugated magnetic nanoparticle suspension failed to reach the necessary temperature for thermal activation, these results point towards a more direct mechanism of energy transfer than simple heat radiation, perhaps via lattice vibrations.

## 4. Materials and Methods

### 4.1. Hydrolysis of SPION to Open Conjugation Sites

In order to maximize induced heating, SPIONs with high specific absorption rates (SAR) (Sigma Aldrich #900042, St. Louis, MO, USA) were used. According to their material safety data sheets, the components of these particles (aside from their iron oxide content) include maleic anhydride polymer with methoxyethane and water [34]. The rate of hydrolysis, prior to laboratory use, of the maleic anhydride to maleic acid is unknown. Thus, we subjected the particles to a rigorous washing in effort to maximize the conversion.

The particle stock solution was first diluted to a working concentration of 0.5 mg/mL, mixed with deionized water, and 190 proof ethanol as a mutual solvent, to a final ratio of 1 part particles, 2 parts water, and 7 parts ethano. The resulting mixture was then heated under reflux conditions for a period of 8 h. Verification of hydrolysis was confirmed via on the NICOLET 6700 FTIR-ATR apparatus (Thermo Fisher Scientific, Waltham, MA, USA).

### 4.2. Carbodiimide Conjugation

Hydrolyzed SPIONs were diluted to a concentration of 0.5 mg/mL. A total 20 mg of both NHS and EDC were added, and the solution was left to incubate for 1 h before the particles underwent magnetic separation. After magnetic separation, the supernatant was decanted and the particles were washed with deionized water. This wash step was repeated three times before the particles were resuspended in PBS and latent TGF-β (2.5 ng/mL) was added. The solution was left to incubate for 1 h before the particles underwent magnetic separation and were washed three times with a PBS solution, wherein the pH was raised to 9 via dropwise addition of NaOH. The particles were washed with pH 9 PBS three times, after which they were resuspended and stored in pH 7 PBS. These wash steps were found to be necessary in order to properly quench the EDC/NHS reaction without altering any existing functional groups, thereby maintaining the integrity of the protein.

### 4.3. Particle Characterization

#### 4.3.1. Fourier Transmission Infrared Attenuated Total Reflectance Spectroscopy (FTIR-ATR)

Liquid samples were placed upright in a lateral magnetic separator (Dobson Laboratory, University of Florida, Gainesville, FL, USA) until the separated magnetic nanoparticles gathered on the lateral side of each upright Eppendorf pipette tube ( Thermo Fisher Scientific, Waltham, MA, USA). After separation, the supernatants were decanted and the particles were resuspended in phosphate buffered saline (PBS). This wash was repeated three times. Prior to use of FTIR-ATR, the particles were sonicated for 10 min in order to ensure a homogenous suspension. A 10 uL aliquot was taken and dried by heating and placed onto the NICOLET 6700 detector (Thermo Fisher Scientific, Waltham, MA, USA). No further preparation was needed, since FTIR-ATR only requires dried-solid samples. A background of air was taken before measurements, and isopropanol was used to clean the detector between samples.

#### 4.3.2. Dynamic Light Scattering (DLS)

The experimental groups were tested against the control groups, all of which were tested against a diluted (1:10) solution of unhydrolyzed SPIONs (0.5 mg/mL). These were measured using a Brookhaven 90Plus Nano Particle Sizer (Brookhaven Instruments, Holtsville, NY, USA ).

#### 4.3.3. Transmission Electron Microscopy

High-resolution transmission electron microscopy (HR-TEM) imaging of the particles was performed using a FEI Tecnai F20 transmission electron microscope operated at 200 kV and equipped with a Gatan UltraScan 1000P digital camera (Thermo Fisher Scientific, Waltham, MA, USA). Specimens for HR-TEM imaging were prepared by drop casting the particles suspended in PBS onto Cu grids with a <2 nm thick C film supported by a thicker, lacy C film (Ted Pella #01824, (Redding, CA, USA), and allowing the grids to dry in air on a warm hotplate. All HR-TEM imaging was performed on particles located on unsupported (<2 nm thick) regions of the C film to minimize the distortion to the images resulting from the presence of the film.

#### 4.3.4. Superconducting Quantum Interference Device (SQUID) Magnetometry

Both experimental and control groups (100 µL of particles in suspension) were collected into PTFE sample holders and placed into the quantum design magnetic property measurement system 3 (superconducting quantum inference device (MPMS-3-SQUID San Diego, CA, USA). Magnetization curves for each group were obtained at room temperature. By fitting the magnetization curve to the Langevin function [34], it is possible to determine the magnetic diameter as well as the geometric deviation; doing this relies on the assumption that the magnetic domains are spherical, as well as the assumption that said magnetic domains possess a magnetization equivalent to bulk magnetite (86.6 emu/g) [35]. We also deduced the saturation magnetization by measuring the maximum magnetization on the magnetization curves [36].

### 4.4. ELISA of Activated TGF-β

#### 4.4.1. Magnetic Release of Active Transforming Growth Factor-β

In order to test the magnetic activation of the latent TGF-β, an enzyme linked immunosorbent assay (ELISA) specific to active TGF-β was used, following the manufacturer’s procedure (Duoset^®^, R&D Systems, Minneapolis, MN, USA). The results were measured via a UV/Vis spectrophotometer (PERKIN-ELMER LAMBDA 800, (Waltham, MA, USA), with absorbance measured at 450 nm. In order to correct for optical imperfections within the plate, absorbance was measured at 540 and 570 nm and subtracted from the 450 absorbance measurement. From the correlation of optical density to protein concentration, a standard curve was created.

Following the conjugation of latent TGF-β to magnetic nanoparticles, magnetic activation by way of magnetic hyperthermia was attempted. First, the particles were magnetically separated and resuspended in PBS. The particles were then split into two groups. The first group was designated for magnetic release. This sub-group would undergo magnetic hyperthermia in a radio frequency alternating magnetic field using an Ambrell Easy Heat (*f* = 350 kHz/H = 40 kA/m) apparatus (Rochester, NY, USA). The second sub-group was used as a control, wherein the samples were kept at room temperature, away from the presence of an alternating magnetic field. For each group, three samples were tested via ELISA, using the given protocol. A two-tailed, unpaired homoscedastic t-test was performed in order to determine significance.

#### 4.4.2. Heat Controls

It became necessary to verify the integrity of the latent TGF-β, and to test whether or not it was possible to release the active TGF-β from the small latent complex using only heat. To this end, latent TGF-β was split into two groups: an experimental group that would be heated on an AccuBlock Digital Dry Bath (Labnet International, Inc., USA) at 80 °C for 10 min, and a control group kept at room temperature. The resulting release of active TGF-β was then quantified via ELISA.

## 5. Conclusions

TGF-β was successfully conjugated to SPIONs using carbodiimide chemistry. Through this research, we have successfully shown the ability to noninvasively transfer energy from a magnetic field to the SPION, triggering the release of active TGF-β from the latent complex. Future studies will require optimization of this technique in order to minimize the background release. Furthermore, we will investigate magneto-mechanical activation strategies in future studies.

## Figures and Tables

**Figure 1 ijms-20-03190-f001:**
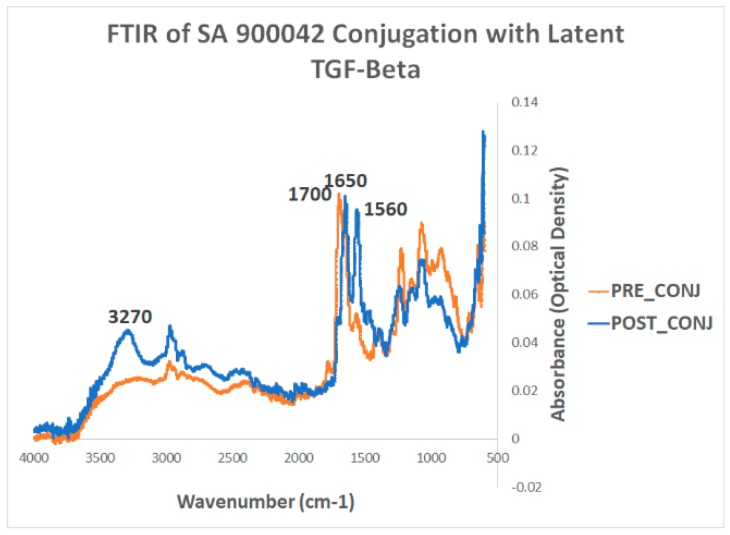
FTIR of Sigma Aldrich 900042 particles before and after conjugation with latent TGF-β. The orange line depicts the “pre-conjugation” absorbance spectrum, whereas the blue line depicts the “post-conjugation” absorbance spectrum.

**Figure 2 ijms-20-03190-f002:**
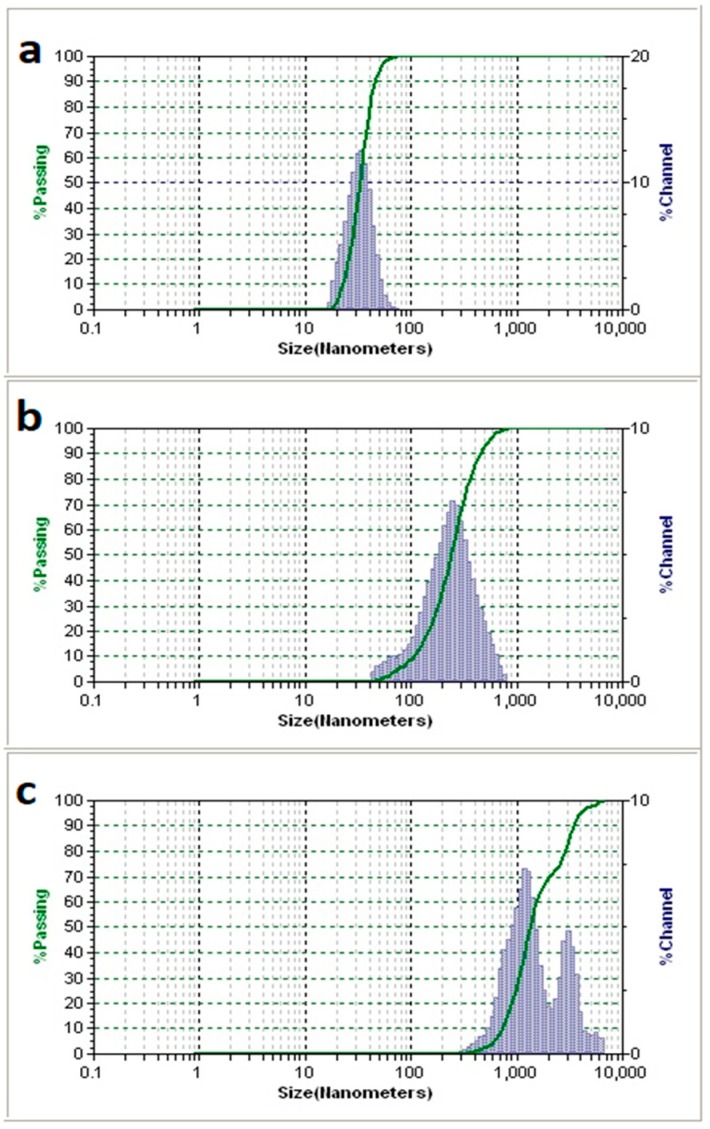
Dynamic light scattering data on the Sigma Aldrich particles #900042. (**a**) depicts the size of the particles alone, (**b**) depicts a measurement of the particles when in solution with NHS as well as latent TGF-β, and (**c**) depicts the size of the particles when conjugated to latent TGF-β with both NHS and EDC.

**Figure 3 ijms-20-03190-f003:**
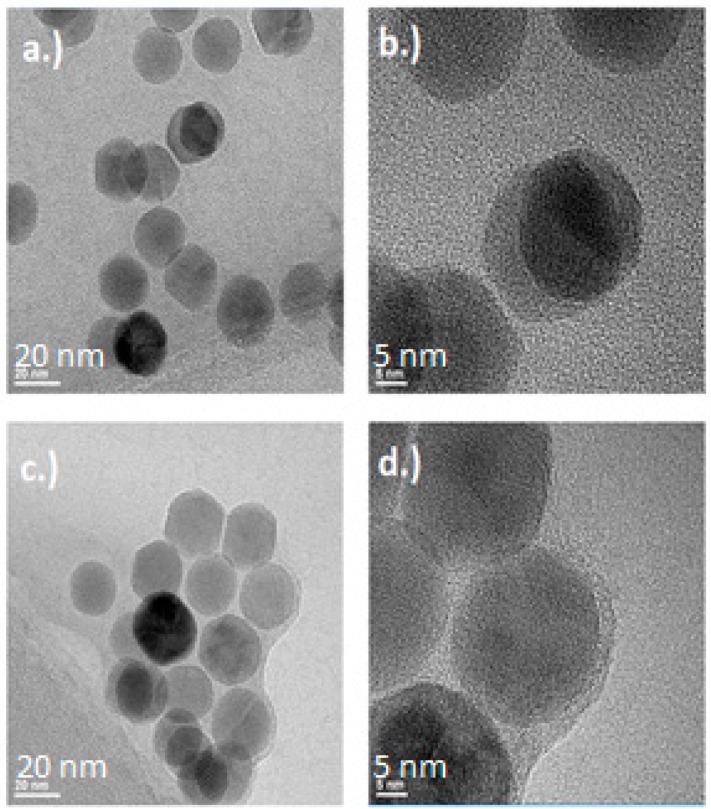
Transmission electron microscope (TEM) images of Sigma Aldrich particle #900042 mixed with latent TGF-β. Images (**a**,**b**) denote the control group, whereas images (**c**,**d**) denote the experimental group. Images (**a**,**c**) are at 145,000× magnification, whereas images (**b**,**d**) are at 400,000× magnification.

**Figure 4 ijms-20-03190-f004:**
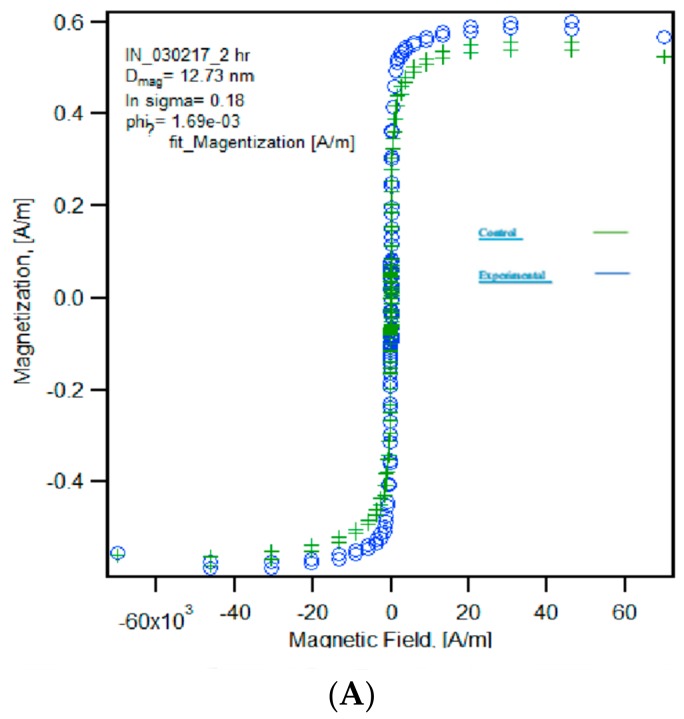
(**A**) Superconducting quantum interference device (SQUID) magnetometer magnetization curves for the magnetic nanoparticles. Green is the control group (particles and protein but lacking in EDC), with saturation magnetization of 2.9 kA/m, whereas blue is the experimental group (particles conjugated to protein), with saturation magnetization of 3.1 kA/m. (**B**) Table depicting Langevin fit model. K0 (temperature) and K1 (magnetization of magnetite) are held variables. K2 is volume fraction, K3 is magnetic diameter, K4 is the geometric deviation, and K5 is the high field slope. Also shown in the table are the number of passes before fit, the chi square of the fit denoting error, and the saturation magnetization.

**Figure 5 ijms-20-03190-f005:**
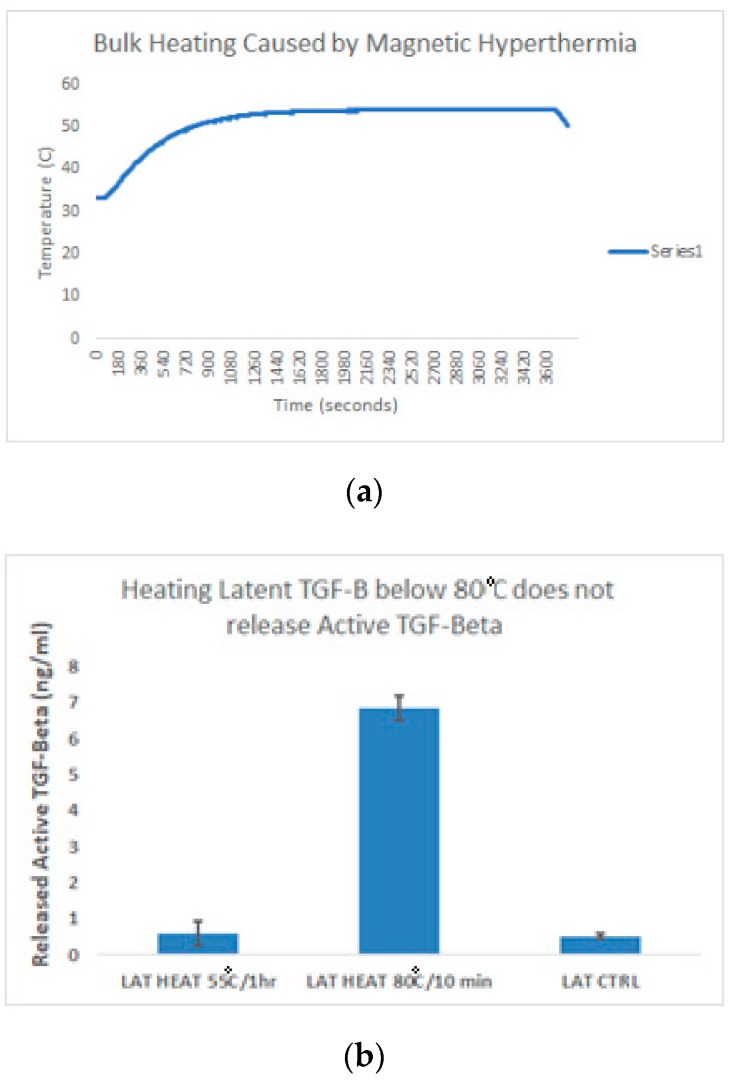
(**a**) Line graph depicting the temperature increase of conjugated particles as a result of magnetic hyperthermia. Magnetic induction began at 120 s, and ended after 3600 s. Maximum temperature is 55 °C. (**b**) Bar graph depicting an ELISA of the release of active TGF-β from unconjugated latent TGF-β due to an increase in bulk temperature.

**Figure 6 ijms-20-03190-f006:**
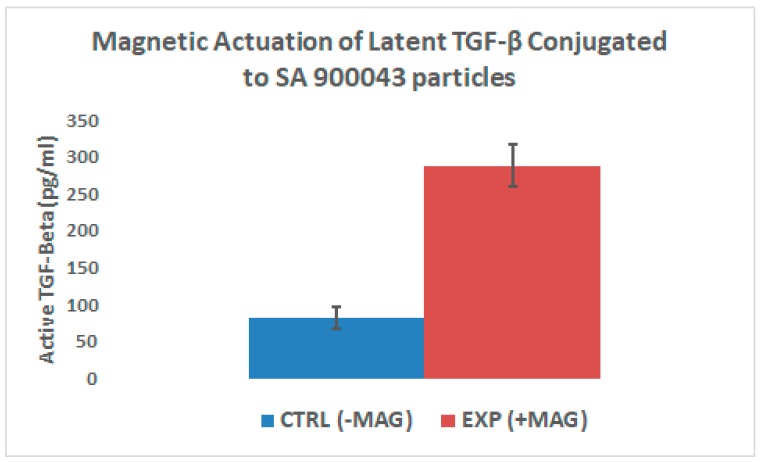
The bar graph depicts two groups of superparamagnetic iron oxide nanoparticles (SPIONs) conjugated to latent TGF-β. The control group was left at room temperature while the experimental group underwent magnetic hyperthermia.

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
