# Peer review of "Carbodiimide Conjugation of Latent Transforming Growth Factor β1 to Superparamagnetic Iron Oxide Nanoparticles for Remote Activation"

_ijms, 2019, doi:10.3390/ijms20133190_

Reviewer 1 Report

The manuscript  is interesting for many researchers and deals with the conjugation of latent transforming growth factor beta1 to superparamagnetic iron oxide. The results are presented clearly. However, the section "Discussion" has to be improved. Now, the discussion sound as a summary but not as a discussion.   

I think that the section "Discussion" should be improved. The authors state that previously the latent form of TGF-beta was conjugated with SPION for the release of active TGF-beta via NIR light activation and via magneto-thermal  activation. In the discussion the authors should explain the differences and advantages of the current work comparing with the previous ones.

Author Response

The discussion has been appropriately transformed from a summary to a discussion of previous and current work within the field, as well as more discussion on the analytical techniques and tests

Reviewer 2 Report

1. Authors should unify the name TGF-β throughout the text (also in graphs).

2. Authors claim the results indicate a statistically significant increase in the release of active TGF-β from the samples that underwent magnetic hyperthermia compared to background release seen in the control. However, there is no information in Materials and Methods section about the statistical test performed in this study.

3. The description of ELISA should appear in chronological way in Materials and Methods section. Firstly in 4.4.1 paragraph Magnetic Release of Active Transforming Growth Factor-β.

4. The temperature is expressed as C instead of °C throughout the text.

Author Response

1. The name TGF-β has been unified throughout the text.

2. The statistical test performed was a 2-tailed unpaired homoscedastic T-Test and both the Materials and Methods section as well as the Results section have been appropriately updated.

3. The paragraphs of 4.4.1 and 4.4.2 of the ELISA section have been appropriately rearranged and includes a description of ELISA.

4. The temperature is appropriately expressed as °C now throughout the text.

Round  2

Reviewer 1 Report

The manuscript has been  significantly  improved and now it is suitable for the publication in present form. 

Reviewer 2 Report

I don't have any comments.